# Modification of Breakfast Fat Composition Can Modulate Cytokine and Other Inflammatory Mediators in Women: A Randomized Crossover Trial

**DOI:** 10.3390/nu15173711

**Published:** 2023-08-24

**Authors:** Jessica M. Delgado-Alarcón, Juan José Hernández Morante, Juana M. Morillas-Ruiz

**Affiliations:** 1Department of Food Technology and Nutrition, Universidad Católica de Murcia, Campus de Los Jerónimos, Guadalupe, 30107 Murcia, Spain; jmdelgado@alu.ucam.edu; 2Eating Disorders Research Unit, Universidad Católica de Murcia, Campus de Los Jerónimos, Guadalupe, 30107 Murcia, Spain

**Keywords:** fatty acids, breakfast, margarine, olive oil, butter, cytokine, inflammation

## Abstract

Previous trials have demonstrated that modifying dietary fat composition can influence the production of inflammation-related factors. Additionally, it has been suggested that not only the type of fat, but also the timing of fat intake can impact these factors. Therefore, the objective of the present study was to evaluate the effect of altering breakfast fat composition on inflammatory parameters. A 3-month crossover randomized trial was designed, involving 60 institutionalized women who alternately consumed a breakfast rich in polyunsaturated fatty acids (PUFA) (margarine), monounsaturated fatty acids (MUFA) (virgin olive oil), or saturated fatty acids (SFA) (butter), based on randomization. The following inflammatory markers were evaluated: epidermal growth factor (EGF), interferon (IFN)-α, interleukin (IL)-1α, IL-1β, IL-2, IL-4, IL-6, IL-8, IL-10, monocyte chemoattractant protein-1 (MCP-1), tumor necrosis factor (TNF)-α, C-reactive protein (CRP), and vascular/endothelial growth factor (VEGF). The results showed that the most significant effects were observed with the high-MUFA breakfast, as there was a statistically significant decrease in plasma IL-6 (*p* = 0.016) and VEGF values (*p* = 0.035). Other factors, such as IL-1α and CRP, also decreased substantially, but did not reach the statistically significant level. On the other hand, the high-PUFA breakfast induced a significant decrease in EGF levels (*p* < 0.001), whereas the high-SFA breakfast had no apparent effect on these factors. In conclusion, modifying breakfast fat, particularly by increasing MUFA or PUFA intake, appears to be sufficient for promoting a lower inflammatory marker synthesis profile and may be beneficial in improving cardiovascular complications.

## 1. Introduction

Although inflammation is a physiological process that helps maintain homeostasis after various stimuli, such as injury or infection [1], several diseases, including obesity and cardiovascular disease-related outcomes, are characterized by impaired chronic low-grade inflammation, which may play a specific role in their pathogenesis [2].

A pro-inflammatory program is activated early in adipose expansion and during obesity, continuously altering the immune system to a proinflammatory phenotype. This phenotype is characterized by the overexpression of proinflammatory cytokines, such as interleukin (IL)-1, IL-6, and tumor necrosis factor-α (TNF-α), along with other acute phase reactants, such as C-reactive protein (CRP), which exceeds the release of anti-inflammatory cytokines, such as IL-4 and IL-10 [3].

Dietary fat, as the main component of cell membranes, has been proposed as a potential link between obesity development and cytokine production. In fact, membrane-derived fatty acids and other related compounds may promote or inhibit cytokine synthesis via the modulation of nuclear factor kappa-B (NF-κB) and/or peroxisome proliferator-activated receptor (PPAR)-γ/α [4,5]. A key point in this argument came from the PREDIMED study, which showed that supplementation with monounsaturated fatty acids (MUFA) produced a greater reduction in pro-inflammatory factors than did a low-fat diet [6,7]. Therefore, the quality of dietary fat appears to be one of the most important factors in regulating the synthesis of these inflammation-related factors.

In agreement with these findings, previous observational studies have also shown that the Mediterranean diet, characterized by high MUFA acid intake, has been associated with a lower degree of inflammation biomarkers, as well as a protective role against cardiovascular events [8]. On the other hand, there is some evidence regarding the pro-inflammatory effects of dietary saturated fatty acids (SFA) [4]. It has been speculated that these fatty acids may induce the expression of inflammatory pathways, including toll-like receptors, protein kinase C, and NOD-like receptors, among others [9]. Polyunsaturated fatty acids (PUFA), specifically the n-6 family, have also been linked to a decrease in pro-inflammatory markers such as IL-6, TNFα, and vascular cell adhesion molecule 1 (VCAM-1) [10]; however, there is some controversy in this regard, as other studies have found the opposite [11].

Recently, new studies have focused not only on the quantity, but also on the time of nutrient intake [12]. The consensus is that eating the greatest number of calories and nutrients at midday is associated with a lower risk of being overweight/obese, whereas a higher intake in the later hours of the day is associated with a higher risk [13,14]. A growing body of chronobiological evidence has highlighted the relevance of breakfast as the most important meal of the day in regards to controlling metabolic homeostasis. Humans are biologically adapted to be awake, active, eating, and storing energy for approximately two-thirds of a 24 h day. During the day, the food we eat provides energy to support metabolic processes, while at night, when people normally sleep, stored energy is mobilized to maintain homeostasis. Thus, inappropriate timing of eating can have desynchronized effects on the circadian clock, leading to adverse health outcomes, such as weight gain, obesity, and poor metabolic health [15]. From an evolutionary perspective, eating high-energy foods in the evening is itself associated with impaired metabolism homeostasis. Thus, repeated and prolonged nighttime eating can lead to circadian disruption, increasing the risk of metabolic syndrome, insulin resistance, and cardiometabolic disease [16]. Other data indicated a potential molecular mechanism by which breakfast skipping induces abnormal lipid metabolism, which is related to the altered circadian oscillation of hepatic gene expression [17].

An interesting work by Jakubowicz et al. showed that simply modifying the energy intake at breakfast was enough to induce greater weight loss and waist circumference reduction in overweight/obese subjects [18]. Konishi et al. also showed that serum concentrations of triglycerides and SFA were significantly decreased in a group of participants with a high intake of fish oil at breakfast [19]. In addition, other studies have shown a better health status in people with a higher protein intake at breakfast [20]. In addition, the recent study by Truman et al. showed that women whose breakfast, as opposed to lunch or dinner, contributed the greatest percentage to their daily fat intake had higher inflammatory marker levels compare to women whose dinner contributed the most to their fat intake [21]. Consequently, as shown by these authors, manipulating breakfast fat composition may be a simple approach to modify the cytokine synthesis profile and, therefore, to obtain a better health status. Unfortunately, to our knowledge, no previous trials have studied this issue.

Therefore, the present work was conducted to elucidate whether changing the fat composition at breakfast can modify cytokine and other inflammatory marker synthesis. Through the present randomized crossover trial, we propose to identify the optimal breakfast fat composition for producing an improved profile of inflammatory markers in women.

## 2. Materials and Methods

### 2.1. Design

To evaluate the effect of breakfast fat composition on cytokine and other inflammation-related factors, a prospective, randomized cross-over trial was conducted. The design employed in the present research, which focused on modifying breakfast to enhance cardiovascular health within a controlled setting, stands out as particularly robust due to its adept handling of confounding factors. By conducting the study within a controlled setting, the work gained a level of control over various environmental variables that could otherwise introduce noise and uncertainty into the results. This setting enables the attenuation of potential confounders, such as dietary habits influenced by communal lifestyle factors, and consistent adherence to a shared routine. Consequently, the findings are poised to provide a clearer and more accurate reflection of the impact of a modified breakfast on inflammation markers, bolstering the internal validity of the present research.

Participants received three different breakfasts: a high-PUFA breakfast (B1: containing 20 g of margarine with no trans fatty acids), a high-SFA breakfast (B2: containing 20 g of butter), and a high-MUFA breakfast (B3: containing 20 g of virgin olive oil). Detailed nutrient analysis of the three breakfasts is provided in Appendix A.

The researchers provided the breakfasts to the participants for 30 consecutive days. Subsequently, a 45-day washing-out period was implemented. The subjects were randomized to partake of one of the three breakfasts, and the type of intervention was alternated, according to a Latin square sequence. The randomization was carried out with the assistance of the Microsoft Excel^®^ program, with a dedicated add-in designed for this purpose.

After obtaining authorization from the Ethics Committee of the Catholic University of Murcia (N # 1874), the patients were informed both verbally and in writing about the study’s design and purpose. They were also provided with an explanation of the ethical aspects of the project, including the main study objective and assurances of their ability to withdraw from the study at any time. Anonymity and the confidentiality of data were also guaranteed. The study adhered to the ethical principles outlined in the Declaration of Helsinki and Spanish legislation on biomedical research. Signing the informed consent was a prerequisite for participation. This trial was registered with the Australian New Zealand Clinical Trials Registry (ACTRN) under number #12611000456954 (detailed information available at: http://www.ANZCTR.org.au/ACTRN12611000456954.aspx, accessed on 23 February 2023).

### 2.2. Participants

Initially, 60 women, aged 64 ± 18 years, with a BMI of 27.79 ± 3.97 kg/m^2^, provided their consent to participate in the study. To ensure similar dietary intake and treatment adherence, the target population consisted of women residing in the same institution. This approach allowed us to ensure uniform dietary and physical activity habits among all participants, as previously published [22]. The exclusion criteria were defined as follows: individuals with cognitive decline, psychiatric disorders, or chronic pharmacological therapy (such as fibrates, statins, other lipid-lowering drugs, oral antidiabetic drugs, thyroid or corticosteroid hormones) that might interfere with the effectiveness of the dietary intervention. Participants who experienced an episode of acute illness or had a pre-existing chronic illness before the study initiation were also excluded. Women undergoing hypocaloric dietary treatment at the time, or within the three months prior to the study, were also excluded.

Sample size calculations were performed using G.Power 3.0 software (Dusseldorf, Germany) [23]. The sample size was determined with a confidence level of 95% and a power of 80%. The sample size was calculated to be sufficient to detect a difference in effect between groups or an effect size (d) corresponding to a 5% change in cytokine or inflammatory marker levels. The a priori standard deviation was derived from our previous study [12]. The program indicated an initial required sample size of 41 subjects in each group. However, considering a potential dropout rate of 5%, a final minimum sample size of 45 subjects per group was calculated. The flow diagram illustrating the progression of the study participants throughout the research is depicted in Figure 1.

### 2.3. Intervention

In addition to the fat source (B1: 20 g of margarine, B2: 20 g of butter, and B3: 20 g of virgin olive oil) (Appendix A), each breakfast included instant coffee (one monodose sachet of 1.8 g), sugar (one monodose sachet of 8 g), and two pieces of white bread toast (60 g). During the intervals between interventions (wash-out period), breakfast consisted of pineapple juice (200 mL) and peach jam (50 g). Except for the type of fat used in the breakfast, the nutritional composition of the other daily meals remained consistent (1636 ± 527 Kcal/day, 61 ± 23 g proteins/day, 203 ± 59 g carbohydrates/day, 65 ± 35 g fats/day). The diets were designed depending on the volunteers’ requirements and based on the volunteer’s nutritional habits in order to enhance adherence. At the start of the study, the volunteers were instructed to follow the assigned diet without altering their lifestyle (physical activity, sleep patterns, meal timings, etc.) during the experimental period.

Apart from the energy and nutrients provided by breakfast, participants adhered to a balanced diet totaling 1636 ± 527 Kcal/day, with a macronutrient distribution of approximately 55 ± 5% carbohydrates, 18 ± 4% protein, and 34 ± 4% fat. Lunch and dinner plans were designed according to participants’ daily energy expenditure and their dietary preferences. An introductory meeting was conducted with the participants at the beginning of the study, emphasizing the importance of maintaining their usual lifestyle habits (eating, physical activity, sleep, etc.) throughout the study duration.

### 2.4. Measurements

Blood samples were collected at the beginning and end of each of the three breakfast interventions, following an overnight fast. Blood was drawn from the antecubital vein into 9 mL tubes. Following venipuncture, the samples were kept on ice and subsequently centrifuged at 10,000× *g* rpm (Heraeus Biofuge Stratos, Thermo Scientific, Dreieich Germany) for 10 min at 4 °C. The mean time between venipuncture and centrifugation was 10 min. Serum samples were then stored at −80 °C until analysis.

Cytokines (interferon (IFN)-γ, interleukin (IL)-1α, IL-1β, IL-2, IL-4, IL-6, IL-8, IL-10, monocyte chemoattractant protein-1 (MCP-1), and tumor necrosis factor (TNF)-α), as well as growth factor vascular/endothelial growth factor (VEGF) and epidermal growth factor (EGF) were analyzed using an Evidence Investigator TM Cytokine I High-Sensitivity Array (following the protocol provided by Randox Laboratories Ltd., Crumlin, UK) utilizing a sandwich immunoassay technique. In brief, 25 µL of serum samples were applied to the biochip and incubated overnight under stable temperature conditions in a thermoshaker. After incubation, the biochips were manually inserted into the image station, and the light signal generated by each analyte was simultaneously detected using digital imaging software provided by the manufacturer. The sensitivity of each analyte, measured by the intra-assay coefficient of variation, was as follows: EGF: 4.6%, IFNγ: 12.7%, IL-1α: 9.7%, IL-1β: 8.2%, IL-2: 9.6%, IL-4: 10.7%, IL-6: 12.9%, IL-8: 10.1%, IL-10: 6.3%, MCP-1: 6.8%, TNFα: 10.2%, and VEGF: 7.3%. Additionally, C-reactive protein (CRP) was determined using an immunoassay with an Invitrogen^®^ CRP Human ELISA Kit, with a sensitivity of <10 pg/mL.

### 2.5. Statistical Analysis

Basic descriptive statistics were calculated as mean ± SD, unless otherwise indicated. The normality of the data distribution was assessed using the Kolmogorov–Smirnov test, given the large sample size. This test confirmed the normal distribution of the obtained data, allowing for the use of parametric tests. Specifically, to assess the effect of the interventions on inflammation markers at the beginning and end of each intervention, a pre-specified, repeated measures ANCOVA was conducted. Time (final versus baseline) and intervention (breakfast 1, 2, or 3) were considered as within-subject factors. In this procedure, age, intervention order, and baseline values were treated as covariates, with treatment as a fixed effect. All statistical tests were performed with a significance level set at *p* < 0.050. The analyses were conducted using SPSS statistical software (version 25.0, SPSS Inc., Chicago, IL, USA).

## 3. Results

### 3.1. Baseline Characteristics of the Participants

At the end of the study, 51 participants successfully completed all three intervention periods, and their data were collected, as depicted in Figure 1. Therefore, all statistical analyses detailed in this study pertain to the participants who successfully completed the study. The clinical and anthropometric baseline characteristics are presented in Table 1. The mean body mass index (BMI) indicates that the participants were classified as overweight. Table 1 also provides an overview of the participants’ clinical history. Hypercholesterolemia was the most prevalent clinical condition, with 19 women having previously been diagnosed with elevated circulating cholesterol levels. All of these characteristics were considered in the analyses of intervention effects (through the ANCOVA method).

### 3.2. Effect of Breakfast Fat Composition on Interleukin Synthesis

There were no statistically significant differences among the different interventions regarding IL1A (*p* = 0.161), despite a 12.5% reduction in mean plasma values in the high-MUFA breakfast group. The same situation was observed for IL1B (*p* = 0.964), IL2 (*p* = 0.846), IL4 (*p* = 0.065), IL8 (*p* = 0.285), and IL10 (*p* = 0.229). Data regarding IL6 showed a similar trend to that of IL1A, since the high-MUFA breakfast induced a 24.5% reduction from baseline IL6 levels. In fact, there was a significant effect of the high-MUFA breakfast compared to high-PUFA breakfast (*p* = 0.025) (Figure 2).

### 3.3. Effect of Breakfast Fat Composition on Other Cytokines and Inflammation Markers

In addition to interleukins, the effect of breakfast fat composition on other cytokines and inflammatory markers was studied in this work. In this regard, there was a statistically significant effect of intervention (*p* = 0.002) and time (*p* = 0.035) on VEGF, since both the high-PUFA and high-MUFA breakfasts decreased the plasma VEGF levels, although the post hoc analysis showed that the effect was statistically significant only with the latter intervention (*p* = 0.035) (Figure 3a,b). There was also an intervention effect regarding EGF (*p* < 0.001), but in this situation, while the high-PUFA breakfast significantly decreased EGF values (*p* < 0.001), the high-MUFA breakfast induced the opposite result (*p* < 0.001) (Figure 3c,d).

On the other hand, the effect of the intervention on IFNγ and MCP1 levels was similar to that of VEGF, since both the high-PUFA and the high-MUFA breakfasts decreased these levels, but there was no significant statistical intervention effect (*p* = 0.560 and *p* = 0.180, respectively). the high-SFA breakfast also decreased TFNα levels, but the effect was not statistically significant (*p* = 0.156). Finally, the data regarding CRP levels also showed an intervention effect (*p* = 0.042), mainly due to the decrease observed in the high-MUFA intervention group (Figure 4). The data obtained in all parameters is shown in Appendix A.

## 4. Discussion

The present work was conducted, in the first place, to determine whether a simple modification of the eating pattern, such as the modification of the type of fat consumed at breakfast, was enough to modify the profile of inflammatory markers. Additionally, we aimed to identify the most suitable fat choice for breakfast to improve the synthesis profile of inflammatory factors. Based on the data obtained, it can be stated that while not all inflammation parameters exhibited significantly modifications, certain changes allowed us to confirm the ability of the intervention to modify the inflammatory profile of the subjects.

As early as 2002, Han et al. conducted a crossover trial with a similar intervention duration (30 days), demonstrating that a diet rich in unhealthy fats led to a significantly higher production of IL-6 and TNFα, compared to a diet abundant in n-6 polyunsaturated fat (soybean oil) [25]. More recently, the long-term consumption of a healthful diet characterized by a high MUFA content was linked to reductions in TNFα and metalloproteinase-9 mRNA, indicating the attenuation of postprandial inflammatory states [26]. In contrast, other studies, such as the short-term investigation by Poppitt et al., which evaluated a high-fat test meal, did not induce changes in CRP, IL-6, or TNFα synthesis [27]. Similarly, the study by Schubert et al., involving a 15-day intervention with both n-3 and n-6 fatty acids, did not reveal any effects on IL-8 or TNFα synthesis [28].

Several factors might contribute to the lack of effect observed in these studies. The type and quantity of fat, for instance, can play a role. Studies using low doses of fat, as seen in Schubert et al. [28], may not be sufficient to elicit a notable change in the inflammatory response. Furthermore, the duration of the intervention is pivotal; studies conducted with short-term interventions or test meals (less than 30 days) might not yield statistically significant effects.

However, the influence of other factors cannot be ruled out. As demonstrated by Schüler et al., dietary patterns and genetics can interact to determine plasma concentrations of markers like VEGF by up to 90% [29]. Fortunately, given the design of this study, the influence of these factors can be partially excluded, allowing us to assert that the obtained data primarily resulted from the intervention undertaken.

Focusing on the specific effects resulting from the various interventions performed in this study, our data indicates that the intervention leading to the most significant changes was the high-MUFA breakfast (virgin olive oil, VOO). Specifically, the high-MUFA breakfast induced a statistically significant reduction in IL6 and VEGF levels, along with observed reductions in the synthesis of other factors like IL1α and CRP. The decrease in IL6, along with its downstream product CRP, holds substantial relevance in enhancing atherosclerotic cardiovascular disease management, given that multiple previous studies in humans have shown that elevated levels of these factors correlate with heightened cardiovascular risk [30,31]. Notably, CRP binds to LDL and is present in atherosclerotic plaques, leading to suggestions of its potential causal role in coronary heart disease [32]. Given the perspective that atherosclerosis fundamentally constitutes a chronic inflammatory disorder [33], it is plausible to argue that augmenting the breakfast with VOO could foster an anti-inflammatory profile, potentially contributing to the reduction of cardiovascular disease incidence.

Previous randomized clinical trials that involved modifying whole dietary fat using olive oil (High-MUFA) have yielded results consistent with those in our study. These trials have demonstrated reductions in IL6 and CRP, as well as decreases in other inflammation markers such as VCAM-1, ICAM-1, and E-selectin [6,34,35]. These effects could potentially be linked to a decrease in postprandial NFκB activity, as evidenced by Cruz-Teno et al. in a 12-week trial involving a high-MUFA diet [26]. However, it is worth noting that some studies have shown no significant effects of MUFA on fasting inflammatory and thrombogenic factors [36]. In this regard, it is important to acknowledge that the within-subject variability in postprandial lipemic response is low [37], and the clinical context, such as the presence of metabolic syndrome, can exert a greater influence on the postprandial response of inflammatory markers than the quality or quantity of dietary fat [38].

The impact of the high-MUFA breakfast was not positive across all inflammatory parameters. Interestingly, while VEGF levels decreased, plasma EGF levels increased significantly by the end of the intervention. Both factors play pivotal roles in vasculogenesis and angiogenesis and have been implicated in vascular pathogenesis [39,40]. Therefore, the overall effect of the high-MUFA breakfast on these factors remains uncertain, warranting further studies for additional clarification. Conversely, the effect of the margarine breakfast (high-PUFA) demonstrated more distinct implications for these growth factors. Although the reduction in VEGF levels was marginal, the decrease in EGF levels was significant. The existing literature provides limited data from human intervention studies with similar findings, although in vitro and animal studies have reported decreases in EGF levels [41] and the genes under their downstream regulation [42] after PUFA supplementation. Collectively, considering the present and previous works, it can be concluded that this type of breakfast holds promise in diminishing the incidence of cardiovascular diseases.

Regarding the high-SFA (butter) breakfast, we did not find any significant change in the inflammatory markers, which is partially in contradiction with previous studies that have shown an increase in the synthesis of pro-inflammatory factors with these types of fatty acids [43,44]. Perhaps, the deleterious effect of SFA fat on the development of obesity and cardiovascular diseases is related, to a greater extent, to lipid metabolism than to the inflammatory response, as we previously described [27].

At this time, it would be important to address certain limitations of the study. Firstly, it must be acknowledged that the findings derived from this study can solely be extrapolated to women. Secondly, although we have examined some of the most pertinent inflammatory markers, numerous other factors remain—particularly those related to molecular adhesion—that could offer additional insights into the impact of different fat types. Another constraint that may have limited the obtention of more pronounced outcomes is the relatively modest quantity of fat dispensed to the participants. Nonetheless, the rationale behind this approach is rooted in our aspiration to evaluate the effects of fat quantities similar to those typically consumed during breakfast. Lastly, despite the fact that the statistical analyses were performed by incorporating the participants’ baseline clinical characteristics, we cannot dismiss the potential influence of other factors that were not considered in this study.

## 5. Conclusions

In conclusion, a seemingly straightforward alteration, such as changing the type of fat eaten at breakfast, may be enough to change an individual’s inflammatory profile. Based on the data obtained, it can be affirmed that the optimal fat choice at breakfast would be a high-MUFA fat, primarily attributable to the noted decrease in the synthesis of pro-inflammatory factors such as IL-1a, IL6, VEGF, and CRP. Moreover, a high-PUFA breakfast may also be adequate since, while its influence on most of the parameters studied was modest, a significant decrease in EGF values was observed, which may be interesting in mitigating the severity of cardiovascular diseases.

## Figures and Tables

**Figure 1 nutrients-15-03711-f001:**
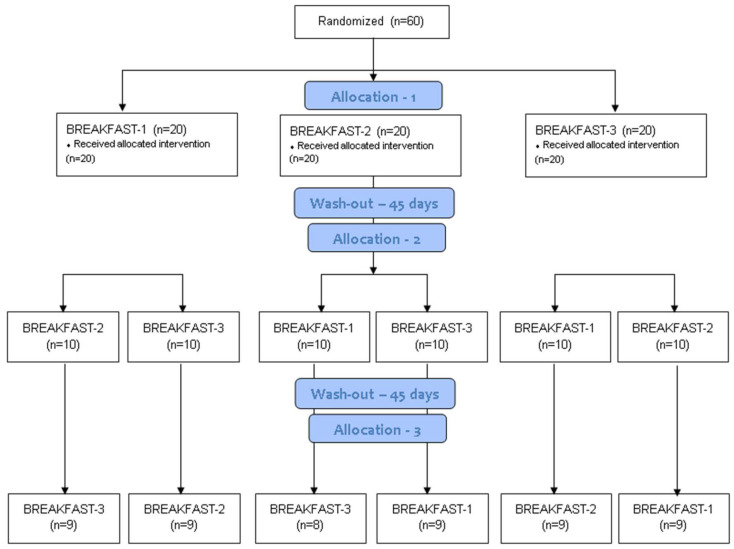
Study flow diagram (this figure has been described elsewhere [24]).

**Figure 2 nutrients-15-03711-f002:**
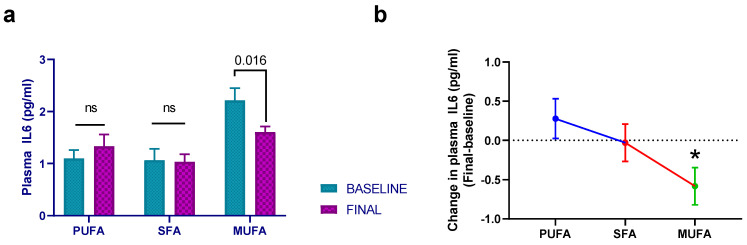
Mean ± SEM plasma IL6 levels at baseline and after 30 days of intervention with the different breakfasts (**a**). (**b**) The estimated treatment differences (final–baseline) in plasma IL6 levels. Data derived from those women who completed the study (*n* = 51). Repeated-measures ANCOVA was employed to evaluate possible statistical differences. Baseline age, BMI, and intervention order were used as covariates. PUFA—margarine-based breakfast; SFA—butter-based breakfast; MUFA—extra virgin olive oil (VOO) breakfast. *—statistically significant differences between the high-MUFA breakfast and the high-PUFA breakfast (*p* = 0.025).

**Figure 3 nutrients-15-03711-f003:**
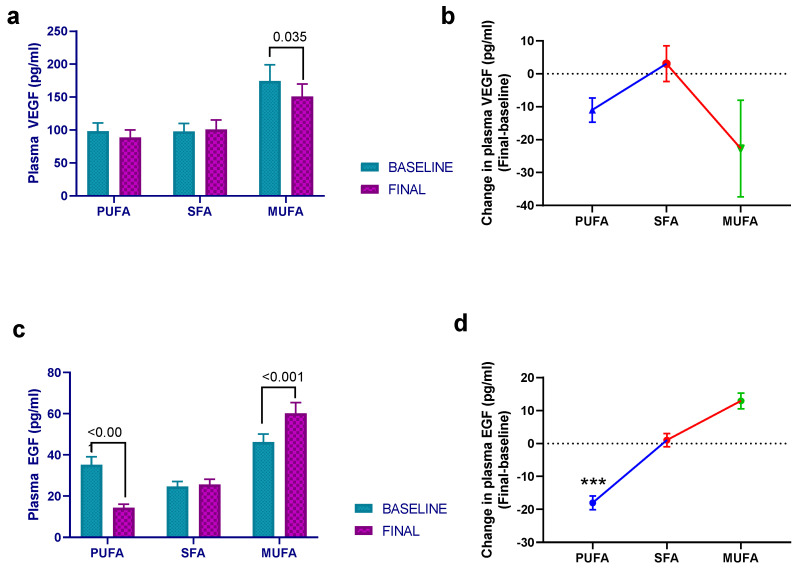
Mean ± SEM plasma VEGF (**a**) and EGF (**c**) levels at baseline and after 30 days of intervention with the different breakfasts. (**b**,**d**) The estimated treatment differences (final–baseline) in plasma levels. Data derived from those women who completed the study (*n* = 51). Repeated-measures ANCOVA was employed to evaluate possible statistical differences. Baseline age, BMI, and intervention order were used as covariates. PUFA— margarine-based breakfast; SFA—butter-based breakfast; MUFA—extra virgin olive oil (VOO) breakfast. ***—statistically significant differences between the high-SFA breakfast and the high-MUFA breakfast (*p <* 0.001).

**Figure 4 nutrients-15-03711-f004:**
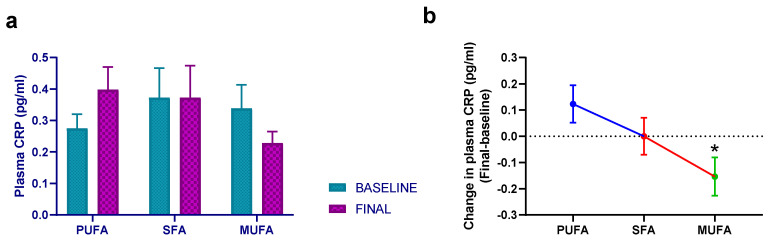
Mean ± SEM plasma CRP levels at baseline and after 30 days of intervention with the different breakfasts (**a**). (**b**) The estimated treatment differences (final–baseline) in plasma CRP levels. Data derived from those women who completed the study (*n* = 51). Repeated-measures ANCOVA was employed to evaluate possible statistical differences. Baseline age, BMI, and intervention order were used as covariates. PUFA—margarine-based breakfast; SFA—butter-based breakfast; MUFA—extra virgin olive oil (VOO) breakfast. *—statistically significant differences between the high-MUFA breakfast and the high-PUFA breakfast (*p =* 0.025).

**Table 1 nutrients-15-03711-t001:** Clinical and anthropometrical baseline characteristics of the study participants.

	Baseline Characteristics (*n* = 53)
Age (years)	63 ± 3 (58–68)
Weight (kg)	64.6 ± 9.0 (62.1–67.2)
BMI	27.77 ± 4.01 (26.63–28.91)
SBP (mmHg)	130 ± 22 (124–136)
DBP (mmHg)	72 ± 9 (69–75)
Hypertension History (*n*; %)	16; 31.4%
Diabetes History (*n*; %)	7; 13.7%
Hypercholesterolemia History (*n*; %)	19; 37.3%
Family History of CVD (*n*; %)	17; 33.3%

Data represent mean ± standard deviation, and the 95% confidence interval is also shown in brackets. BMI: body mass index; SBP: systolic blood pressure; DBP: diastolic blood pressure; CVD: cardiovascular disease.

## Data Availability

The data presented in this study are available on request from the corresponding author. The data are not publicly available because we do not have the explicit approval of the religious association that collaborated in the study.

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
