# Peer review of "Modification of Breakfast Fat Composition Can Modulate Cytokine and Other Inflammatory Mediators in Women: A Randomized Crossover Trial"

_nutrients, 2023, doi:10.3390/nu15173711_

Round 1
Reviewer 1 Report
This is an interesting title where the authors claim that quality of breakfast fat can modulate inflammatory markers. However, I have some concerns regarding the interpretation of data in that the effect could be due to general consumption rather than time specific consumption, as no comparable consumption had been reported for later part of the day. Secondly the authors should also improve the Introduction section by including more studies on chrononutrition of dietary fat intake and why was breakfast chosen in particular as the meal for intervention. I would also prefer to see all the inflammatory markers reported in the graphs rather than leaving out the statistically non-significant ones. For Figure 3b, its also intriguing to notice why the baseline concentration of VEGF was higher compared to the other treatments, particularly considering the crossover trial design.
Overall significant improvements in both English language and interpretation need to be made before this can be accepted for publication
Author Response
NUTRIENTS-2554796 “Modification of breakfast fat composition can modulate cytokine and other inflammatory mediators in women: a randomized crossover trial”.
Reviewer#1
Comments:
This is an interesting title where the authors claim that quality of breakfast fat can modulate inflammatory markers. However, I have some concerns regarding the interpretation of data in that the effect could be due to general consumption rather than time specific consumption, as no comparable consumption had been reported for later part of the day.
To begin with, we express our gratitude for the reviewer's comments. It is possible that the original paper lacked sufficient information regarding this matter. We believe that a key aspect of this paper revolves around the fact that all participants were situated within the same institution. Consequently, uniformity in physical activity and diet was ensured among all participants. Any disparities in dietary intake were kept to a minimum. In light of this, we have introduced additional sentences in the methods section to provide further elucidation on this particular issue.
Secondly the authors should also improve the Introduction section by including more studies on chrononutrition of dietary fat intake and why was breakfast chosen in particular as the meal for intervention.
We completely agree with the reviewer’s comment. We have included the following references about the chrononutrition of dietary fat intake [1,2]. On the other hand, one of the reasons to justify the breakfast meal was the previous work of Jakubowicz et al., where they showed that modification of breakfast energy content was enough to obtain higher weight loss [3]. The present work was carried out, in some way, to confirm that a modest modification as changing breakfast fat content was useful to improve cardiometabolic parameters.
I would also prefer to see all the inflammatory markers reported in the graphs rather than leaving out the statistically non-significant ones.
The comment of the reviewer is fairly interesting, and it would be an important point to fully understand the data obtained. Nevertheless, we hold the view that incorporating an excessive number of figures might divert attention from the primary observed effects. Nonetheless, we have provided all discerned changes in the studied parameters as supplementary material. Should the reviewer deem it indispensable, they are welcome to include the detailed data in a subsequent review."
For Figure 3b, its also intriguing to notice why the baseline concentration of VEGF was higher compared to the other treatments, particularly considering the crossover trial design.
Indeed, the reviewer's comment is quite insightful. This scenario intrigued us as well. In fact, this pattern was also observed in some parameters, though not across all of them. It's possible that certain parameters experienced seasonal variations, but the revised bibliography lacks information in this regard. Nevertheless, we maintain that the primary focus of this study was to contrast baseline and final data, as well as to compare absolute changes among the groups. As a result, the statistical procedure employed to assess these changes was an ANCOVA adjusted for baseline values, as outlined in the original version of the paper."
Comments on the Quality of English Language
Overall significant improvements in both English language and interpretation need to be made before this can be accepted for publication.
We regret the various grammatical mistakes that were present. These errors have been rectified through professional editing services. We would like to emphasize that several changes made to the revised paper have not been explicitly highlighted. This approach was taken to prevent any potential confusion and enhance the overall clarity of the review.
In summary, we would like to thank the reviewer’s comments. We considered that the changes made following the indications have significantly improved the quality of the paper.
Juan José Hernández Morante on behalf of all my co-authors.
Reviewer 2 Report
Modification of breakfast fat composition can modulate cytokine and other inflammatory mediators in women: a randomized crossover trial; Delgado-Alarcón et al.
The paper presented by Delgado-Alarcón et al. investigates how changing the fat type consumed daily at breakfast will impact systemic inflammatory mediators to mitigate predecessors of cardiovascular conditions. As a whole the study provides an interesting look at how a small dietary change may impact overall systemic parameters, though significance was lacking overall.
Comments:
1. Please include the nutrient analysis for each breakfast.
2. Were participants instructed to eat all of the breakfast each day? Were there variations in consumption among the groups? Further, while all participants were offered the same meals as they were institutionalized, was intake verified and were there any deviations that may impact results?
3. Baseline measures of groups were provided, were these repeated following each portion of the study? For example, it would be important to show any changes in anthropometrics following each intervention.
4. As authors are suggesting the inflammatory profile is linked to cardiovascular disease, authors should discuss the cytokines presented and their mechanism of action as it relates to cardiovascular conditions.
5. Many issues throughout the text were found in regards to the use of consistent abbreviations. Authors should revise accordingly to be sure all cytokines and other parameters are defined prior to use. All abbreviations should have consistent nomenclature. Further, authors should be consistent in how they refer to the diets to avoid reader confusion. For example, discuss the diets based on either their type of fat (i.e., MUFA, PUFA) or source (i.e., margarine, olive oil) consistency throughout the manuscript.
6. Data in the bar groups of 2 and 3 should be presented as mean ± standard deviation for accurate representation of the data.
7. Revise figure labeling so that each graph (bar or line) has its own label. For example, in Figure 2, label the bar graph as A and the line graph as B.
Author Response
NUTRIENTS-2554796 “Modification of breakfast fat composition can modulate cytokine and other inflammatory mediators in women: a randomized crossover trial”
Reviewer#2
The paper presented by Delgado-Alarcón et al. investigates how changing the fat type consumed daily at breakfast will impact systemic inflammatory mediators to mitigate predecessors of cardiovascular conditions. As a whole the study provides an interesting look at how a small dietary change may impact overall systemic parameters, though significance was lacking overall.
Comments:
- Please include the nutrient analysis for each breakfast.
This information has been included as supplementary table S1.
- Were participants instructed to eat all of the breakfast each day? Were there variations in consumption among the groups? Further, while all participants were offered the same meals as they were institutionalized, was intake verified and were there any deviations that may impact results?
This point is a good remark by the reviewer. Effectively, as being institutionalized, all participants eat the same food. The dietary regimen was exactly identical for all participants, following the usual procedure of the institution. Physical activity was also similar. Variations in intake were minimal, as verified by the researchers in the study.
- Baseline measures of groups were provided, were these repeated following each portion of the study? For example, it would be important to show any changes in anthropometrics following each intervention.
Again, this comment is quite interesting. In a previous study, conducted on the same volunteers, described changes on several metabolic parameters, although no changes in anthropometrical characters were observed. However, as these results were previously reported [1], we wanted to focus on the effect on cytokines and inflammation markers.
- Delgado-Alarcón, J.M.; Morante, J.J.H.; Aviles, F. V.; Albaladejo-Otón, M.D.; Morillas-Ruíz, J.M. Effect of the Fat Eaten at Breakfast on Lipid Metabolism: A Crossover Trial in Women with Cardiovascular Risk. Nutrients 2020, 12, 1–13, doi:10.3390/nu12061695.
- As authors are suggesting the inflammatory profile is linked to cardiovascular disease, authors should discuss the cytokines presented and their mechanism of action as it relates to cardiovascular conditions.
We completely agree with the reviewer and regret the lack of information. We have included further information about the mechanism of action of the parameters on the discussion section. The following references have also been included: [2–5]
- Cagnina, A.; Chabot, O.; Davin, L.; Lempereur, M.; Maréchal, P.; Oury, C.; Lancellotti, P. Atherosclerosis--an Inflammatory Disease. N. Engl. J. Med. 1999, 340, 302–309, doi:10.1056/NEJM199901143400207.
- Liu, W.; Yalcinkaya, M.; Maestre, I.F.; Olszewska, M.; Ampomah, P.B.; Heimlich, J.B.; Wang, R.; Vela, P.S.; Xiao, T.; Bick, A.G.; et al. Blockade of IL-6 Signaling Alleviates Atherosclerosis in Tet2-Deficient Clonal Hematopoiesis. Nat. Cardiovasc. Res. 2023, 2, 572, doi:10.1038/S44161-023-00281-3.
- Kaptoge, S.; Di Angelantonio, E.; Lowe, G.; Pepys, M.B.; Thompson, S.G.; Collins, R.; Danesh, J.; Tipping, R.W.; Ford, C.E.; Pressel, S.L.; et al. C-Reactive Protein Concentration and Risk of Coronary Heart Disease, Stroke, and Mortality: An Individual Participant Meta-Analysis. Lancet 2010, 375, 132, doi:10.1016/S0140-6736(09)61717-7.
- Ridker, P.M.; Rifai, N.; Stampfer, M.J.; Hennekens, C.H. Plasma Concentration of Interleukin-6 and the Risk of Future Myocardial Infarction among Apparently Healthy Men. Circulation 2000, 101, 1767–1772, doi:10.1161/01.CIR.101.15.1767.
- Many issues throughout the text were found in regards to the use of consistent abbreviations. Authors should revise accordingly to be sure all cytokines and other parameters are defined prior to use. All abbreviations should have consistent nomenclature. Further, authors should be consistent in how they refer to the diets to avoid reader confusion. For example, discuss the diets based on either their type of fat (i.e., MUFA, PUFA) or source (i.e., margarine, olive oil) consistency throughout the manuscript.
We regret the lack of consistency of both abbreviations and diet nomenclature. As suggested by the reviewer, we have unified diet nomenclature based on fatty acid composition. In addition, abbreviations of cytokines and other parameters were stated when first defined.
- Data in the bar groups of 2 and 3 should be presented as mean ± standard deviation for accurate representation of the data.
Precisely, the use of standard error was used to better represent the data obtained in the present work. Perhaps, the use of SD is more standardized that SEM. However, several cytokine data showed a great dispersion, and in our opinion, the use of SEM clarify the interpretation of the data obtained. Furthermore, the use of sem has been previously used by our research group [6,7] in previous works and it is widespread used in the bibliography [8].
- Hernandez-Morante JJ, Cerezo D, Cruz RM, Larque E, Zamora S, Garaulet M. Dehydroepiandrosterone-sulfate modifies human fatty acid composition of different adipose tissue depots. Obes Surg 2011; 21(1):102-11. (Table 1).
- Hernández Morante JJ, Sánchez-Villazala A, Cañavate Cutillas R, Conesa Fuentes MC. Effectiveness of a Nutrition Education Program for the Prevention and Treatment of Malnutrition in End-Stage Renal Disease. J Ren Nutr 2014; 24(1):42-9. doi: 10.1053/j.jrn.2013.07.004. (Table 1)
- Matikainen N, Björnson E, Söderlund S, Borén C, Eliasson B, Pietiläinen KH, Bogl LH, Hakkarainen A, Lundbom N, Rivellese A, Riccardi G, Després JP, Alméras N, Holst JJ, Deacon CF, Borén J, Taskinen MR. Minor Contribution of Endogenous GLP-1 and GLP-2 to Postprandial Lipemia in Obese Men. PLoS One. 2016;11(1):e0145890. (Figure 1)
Nevertheless, if the reviewer still considers it necessary, we will modify the figures accordingly.
- Revise figure labeling so that each graph (bar or line) has its own label. For example, in Figure 2, label the bar graph as A and the line graph as B.
Again, we regret these mistakes. We have modified the figures accordingly.
In summary, we would like to thank the reviewer’s comments. We considered that the changes made following the indications have significantly improved the quality of the paper. We would like to clarify that several abbreviations and the unification of breakfast composition have not been highlighted to better observed the other comments.
Juan José Hernández Morante on behalf of all my co-authors.
Round 2
Reviewer 1 Report
I am afraid the authors have fully addressed my previous comments in that the study design doesn't address the effects of dietary fat manipulation in breakfast per se and it can just be the general dietary change that was introduced? This must be acknowledged appropriately.
Some improvement
Author Response
We would like to express our regret for our inability to adequately address all the concerns raised in your previous review. We acknowledge that some of the points you raised were indeed crucial and could have added substantial value to our paper. However, we assure you that we have carefully considered your comments and we have tried to significantly modify the paper based on the suggestions provided.
Nevertheless, we understand your concern about the adequacy of the employed methodology. There are several points that justify the methodology employed in the present study.
On the one hand, we would like to highlight that the methodology employed in our study was chosen based on its proven effectiveness and relevance in previous works [1,2]. As you correctly pointed out, the methodology used in our study has been previously employed in other works with successful outcomes. This established track record provides confidence in its suitability for our research objectives and allows for meaningful comparisons between studies.
- Keogh, G.F.; Cooper, G.J.S.; Mulvey, T.B.; McArdle, B.H.; Coles, G.D.; Monro, J.A.; Poppitt, S.D. Randomized controlled crossover study of the effect of a highly β-glucan-enriched barley on cardiovascular disease risk factors in mildly hypercholesterolemic men. Am. J. Clin. Nutr. 2003, 78, 711–718, doi:10.1093/ajcn/78.4.711.
- Samkani, A.; Skytte, M.J.; Anholm, C.; Astrup, A.; Deacon, C.F.; Holst, J.J.; Madsbad, S.; Boston, R.; Krarup, T.; Haugaard, S.B. The acute effects of dietary carbohydrate reduction on postprandial responses of non-esterified fatty acids and triglycerides: A randomized trial. Lipids Health Dis. 2018, 17, 1–9, doi:10.1186/s12944-018-0953-8.
On the other hand, Regarding the study population, we agree that the context in which the research takes place is of paramount importance. Our study was conducted at an institution that has maintained a consistent dietary pattern over the years. This aspect offers a unique advantage in ensuring a controlled and stable environment for dietary interventions. Unlike free-living or community populations, the stability and uniformity of the institutional dietary pattern contribute to reducing potential confounding factors that might arise from varying lifestyles and dietary habits. The decision to utilize this specific population was deliberate, as it aligns with the research focus and aims to provide insights into the specific effects of the intervention without the noise of external factors. This approach enhances the internal validity of our findings and strengthens the credibility of our conclusions. Again, previous works validates the present methodology [3-6]:
- Jones DW, Luft FC, Whelton PK, Alderman MH, Hall JE, Peterson ED, Califf RM, McCarron DA. Can We End the Salt Wars With a Randomized Clinical Trial in a Controlled Environment? Hypertension. 2018;72(1):10-11. doi: 10.1161/HYPERTENSIONAHA.118.11103.
- de Mello VD, Dahlman I, Lankinen M, Kurl S, Pitkänen L, Laaksonen DE, Schwab US, Erkkilä AT. The effect of different sources of fish and camelina sativa oil on immune cell and adipose tissue mRNA expression in subjects with abnormal fasting glucose metabolism: a randomized controlled trial. Nutr Diabetes. 2019;9(1):1. doi: 10.1038/s41387-018-0069-2.
- Cortie CH, Byrne MK, Collier C, Parletta N, Crawford D, Winberg PC, Webster D, Chapman K, Thomas G, Dally J, Batterham M, Martin AM, Grant L, Meyer BJ. The Effect of Dietary Supplementation on Aggressive Behaviour in Australian Adult Male Prisoners: A Feasibility and Pilot Study for a Randomised, Double Blind Placebo Controlled Trial. Nutrients. 2020;12(9):2617. doi: 10.3390/nu12092617.
- Derose KP, Williams MV, Flórez KR, Griffin BA, Payán DD, Seelam R, Branch CA, Hawes-Dawson J, Mata MA, Whitley MD, Wong EC. Eat, Pray, Move: A Pilot Cluster Randomized Controlled Trial of a Multilevel Church-Based Intervention to Address Obesity Among African Americans and Latinos. Am J Health Promot. 2019;33(4):586-596. doi: 10.1177/0890117118813333.
We have incorporated your valuable feedback into our revised manuscript, emphasizing the rationale behind our choice of methodology and study population. We believe that these clarifications will help address the concerns you raised and provide a more comprehensive understanding of the study's design.
Once again, we express our gratitude for your thoughtful review and constructive comments. Your input has undoubtedly contributed to the refinement of our work.